# Temporal Trends of *Escherichia coli* Antimicrobial Resistance and Antibiotic Utilization in Australian Long-Term Care Facilities

**DOI:** 10.3390/antibiotics14020208

**Published:** 2025-02-18

**Authors:** Chloé Corrie Hans Smit, Caitlin Keighley, Kris Rogers, Spiros Miyakis, Katja Taxis, Hamish Robertson, Lisa Gail Pont

**Affiliations:** 1Graduate School of Health, University of Technology Sydney, 100 Broadway, Sydney, NSW 2008, Australia; chloe.smit@uts.edu.au (C.C.H.S.);; 2Graduate School of Medicine, University of Wollongong, Building 28, Wollongong, NSW 2522, Australia; 3Southern.IML Pathology, 3 Bridge St, Coniston, NSW 2500, Australia; 4Department of Infectious Diseases, Lawson House, Wollongong Hospital, Loftus Street, Wollongong, NSW 2500, Australia; 5PharmacoTherapy, -Epidemiology and -Economics—Groningen Research Institute of Pharmacy, University of Groningen, Antonius Deusinglaan 1, 9713 AV Groningen, The Netherlands; 6School of Design, Queensland University of Technology, 149 Victoria Park Road, Kelvin Grove, QLD 4059, Australia

**Keywords:** antimicrobial resistance, *Escherichia coli*, long term care, urinary tract infections

## Abstract

**Background/Objectives:** Antimicrobial resistance (AMR) is a global problem with antibiotic consumption considered a key modifiable factor for the development of AMR. Long-term care (LTC) facilities have been identified as potential reservoirs for *Escherichia coli* (*E. coli*) resistance due to high rates of urinary tract infection (UTI) and high levels of antibiotic consumption among residents. However, while the relationship between these two factors is well accepted, little is known about the possible temporal relationship between these. This study explores trends in *E. coli* resistance and antibiotic consumption in LTC focused on potential temporal relationships between antibiotic utilization and AMR. Methods: A retrospective, longitudinal, and ecological analysis was conducted between 31 May 2016 and 31 December 2018. The primary outcomes were the monthly prevalence of *E. coli* AMR in urine isolates and the monthly percentage of residents using an antibiotic recommended for the management of UTI in national treatment guidelines (amoxicillin, amoxicillin with clavulanic acid, cefalexin, norfloxacin, and trimethoprim). Results: During the study period, 10,835 urine *E. coli* isolates were tested, and 3219 residents received one or more medicines and were included in the medicines dataset. Over one-quarter were resistant to at least one of the target antibiotics (23.3%). For most antibiotics, the temporal relationship between AMR and antibiotic utilization was unclear; however, potential patterns were observed for both trimethoprim and amoxicillin with clavulanic acid. Trimethoprim showed a temporal decrease in both AMR and utilization, while amoxicillin with clavulanic acid showed a lag time of approximately four months between utilization and resistance. Conclusions: The dynamic nature of AMR demonstrated in this study highlights the need for more up-to-date local surveillance to inform antibiotic choice in this setting.

## 1. Introduction

Antimicrobial resistance (AMR) is a global issue and a significant threat to human health. With antibiotic development stagnant, rising AMR has significant consequences including an increase in untreatable infections, extended hospital stays, and increased mortality [1]. Antibiotic utilization has been identified as a key driver of AMR [2] with high antibiotic utilization leading to the depletion of susceptible bacterial strains over time, creating reservoirs of multidrug-resistant bacteria [3].

Long-term care (LTC) facilities have been identified as significant AMR reservoirs [4]. It is proposed that the semi-closed LTC environment increases the spread of in-house organisms between residents, with the transfer of residents between community LTC and hospital settings facilitating the spread of AMR between healthcare contexts [5,6]. Previous research has shown that LTC residents also have high rates of antibiotic utilization [7,8,9,10,11,12], with urinary tract infections (UTI) the most common reason for antibiotic prescription in the LTC setting [13]. The Australian treatment guidelines recommend a range of antibiotics for the treatment of UTI including amoxicillin with clavulanic acid, trimethoprim and cefalexin [14]. A significant consideration for clinicians working in the LTC setting is balancing the benefit of antibiotic utilization for the individual resident against the emergence of AMR at the population level [15].

*Escherichia coli* (*E. coli*), a commensal of the human gastrointestinal system, is responsible for up to 80% of UTIs [16]. The transition of commensal *E. coli* to pathogenic antimicrobial-resistant *E. coli* is mainly driven by the horizontal gene transfer of resistant genes or mobile genetic elements. The mechanism of resistance varies per antibiotic type [17]. Resistance to amoxicillin is mediated via acquisition of the TEM-1/TEM-2- β-lactamase, whereas the presence of extended-spectrum β-lactamase leads to the resistance of *E. coli* to amoxicillin with clavulanic acid and cefalexin. Trimethoprim resistance is established through mutations of *sul* and *dfr* genes, and gyrA/parC and gyrB/pare are responsible for resistance to quinolones [17].

Empirical antibiotic treatment, i.e., prescribing antibiotics prior to microbiological confirmation for UTIs, is common in LTC facilities [18] and based on local AMR patterns. However, AMR research to date is derived from the hospital setting, despite previous research showing that hospital AMR patterns deviate considerably from those in LTC facilities [19]. Furthermore, the seasonality in antibiotic use [20] and the dynamic nature of resistance suggest the need for temporal surveillance of both use and AMR [18].

Obtaining comprehensive longitudinal microbiology and antibiotic utilization data in the LTC setting can be challenging [21]. In many settings, it is not possible to identify LTC residents separately from those residing in the general community within microbiology datasets [22], and antibiotic utilization in LTC is often presented as a point prevalence rather than repeated or longitudinal data points [23,24]. The Illawarra Shoalhaven region, south of Sydney in Australia, provides a unique opportunity to explore the temporal relationship between AMR and antibiotic utilization in the LTC setting. Within the region, a single microbiology laboratory services the region [25], longitudinal data on antibiotic utilization in LTC are available [20], and the Australian long-term care setting is considered comparable to LTC settings across North America, Europe, and Oceania [26].

Antibiotic stewardship is a well-established strategy for mitigating AMR in the hospital setting, and interest in the introduction of antimicrobial stewardship models in LTC globally is growing. Understanding the fundamental time-scale changes in antibiotic utilization and AMR in the LTC setting is important for antibiotic stewardship [27]. Therefore, this study aims to explore temporal trends in AMR and antibiotic utilization in the LTC setting, focusing on *E. coli* resistance in urine isolates and the antibiotics commonly used to manage UTIs.

## 2. Results

### 2.1. Cohort Characteristics

During the study period, 10,835 urine *E. coli* isolates were tested and included in the microbiology dataset, and 3219 residents received one or more medicines and were included in the medicine dataset.

Out of the 10,835 urine *E. coli* isolates included in the dataset, 23.3% (2526/10,835) were resistant to at least one of the five antibiotics recommended for treatment of UTI in the national guidelines. *E. coli* resistance was highest to amoxicillin (51.3%) and trimethoprim (28.6%).

Over the 2.5 years, residents utilized 6795 antibiotic treatment episodes, of which slightly over half were cefalexin (51.2%), followed by trimethoprim at 15.4% and amoxicillin with clavulanic acid at 15.0% (Table 1).

### 2.2. Monthly Trends of E. coli AMR

In the urine isolates tested, the rate of *E. coli* resistance was highest for amoxicillin (560 resistant isolates/1000 isolates tested (95% CI 495–633)) at the start of the study period in June 2016. *E. coli* amoxicillin resistance decreased throughout the study period reaching 457/1000 (95% CI 405–516) at the end of the study period (Figure 1). Trimethoprim had the second highest rate of resistance, with resistance also decreasing throughout the study period (338/1000 isolates resistant (95% CI 287–397) in June 2016 to 234/1000 isolates (95% CI 198–277) in December 2018).

*E. coli* resistance to both cefalexin and amoxicillin with clavulanic acid fluctuated over time. The lowest resistance to cefalexin was measured in February 2017 (89/1000 95% CI 68–118) and the highest in June 2016 (163/1000 isolates 95% CI 104–254) and November and December 2017 (149/1000 CI 118–188). Amoxicillin with clavulanic acid resistance was highest in June 2016 (172/1000 CI 103–289), April 2017 (119/1000 95% CI 84–169), and October 2018 (120/1000 95% CI 88–164) and lowest in September/October 2017 (75/1000 95% CI 51–110) and May/June 2018 (77/1000 95% CI 52–114).

Overall, norfloxacin resistance remained around 111 per 1000 isolates resistant (95% CI 97–129) each month. Decreases in testing for and the utilization of norfloxacin between August 2017 and April 2018 coincided with an international shortage of norfloxacin (Table 1).

### 2.3. Temporal Relationship Between E. coli AMR and Antibiotic Utilization

Similarities in the temporal trends between *E. coli* AMR and antibiotic utilization were observed for two antibiotics: trimethoprim and amoxicillin with clavulanic acid.

A decrease in trimethoprim utilization from June 2017 (24/1000 residents 95% CI 21–28) to December 2018 (15/1000 residents 95% CI 11–21) was followed by a decrease in resistance to trimethoprim during the same period (300/1000 95% CI 274–329 to 234/1000 95% 198–277).

In contrast, increases in the utilization of amoxicillin with clavulanic acid were followed by increases in *E. coli* resistance to amoxicillin with clavulanic acid with a lag time of approximately four to five months. Similarly, increased amoxicillin with clavulanic acid utilization in January 2017 and August 2017 (23/1000 95% CI 20–27 and 22/1000 95% CI 19–26) was followed by an increase in resistance in April 2017 and January 2018, respectively (119/1000 95% CI 84–169 and 107 95% CI 76–150).

No clear trends that coincided were visible for amoxicillin and cefalexin, and there were too few data on norfloxacin to compare the trends visually.

## 3. Discussion

In this study, we showed dynamic trends in *E. coli* resistance and antibiotic utilization. The observed trends appeared antibiotic-specific with a potential temporal relationship observed for both trimethoprim and amoxicillin with clavulanic acid, but no relationship was observed for amoxicillin or cephalexin. We also found high antimicrobial resistance of *E. coli* in LTC facilities, with one in five tested urine isolates resistant to an antibiotic.

In this study, we found possible temporal relationships between AMR and antibiotic utilization for trimethoprim and amoxicillin with clavulanic acid, but not for cephalexin or amoxicillin, and we were unable to assess a possible relationship for norfloxacin due to insufficient data in both the medicine and microbiology datasets. Previous studies in the community have shown a lag time between population antibiotic utilization and AMR development varying from one to six months [28,29,30]; however, our results indicate that this may be antibiotic-specific. We found a possible lag time of four to five months between the utilization of amoxicillin with clavulanic acid and the development of *E. coli* resistance against this antibiotic. This contrasts with other studies that found a 2-month lag [28] or no temporal relation for this antibiotic [31]. Similar to results reported elsewhere, we observed that a decrease in the utilization of trimethoprim was followed by a reduction in resistance [30]. Our findings of a lack of temporal relationship in the LTC also align with findings from community settings [28,31], where it has been proposed that cefalexin utilization in hospitals could directly drive community-level resistance [31].

The dynamic trends of resistance patterns of *E. coli* observed in this study in the LTC setting align with findings from previous research into AMR in the community setting [27,28,30,32,33]. Overall, we found a slight decrease in antibiotic utilization and *E. coli* AMR aligning with previous research investigating urine *E. coli* isolates after antibiotic utilization reductions [34,35]. Previous research proposed that maintaining antimicrobial resistance depends on the volume of antibiotics utilized [36] and the biological fitness cost, described as strains having to divert some of their limited energy from reproduction to maintaining their antimicrobial resistance traits over time [37,38]. The effect of this is that at a lower level of antibiotic utilization, susceptible bacteria will regain a natural reproductive edge over resistant strains, leading to a reversal of resistance [37].

We observed relatively high rates of *E. coli* resistance for both trimethoprim and amoxicillin. We found that 28.6% of *E. coli* isolates were resistant to trimethoprim, the first choice for empirical treatment according to the Australian guidelines [14]. The choice of empirical treatment, i.e., treatment without confirmed pathology results, is based on local pathology susceptibility results [18,39]. There is no universal clinical susceptibility threshold for antibiotic choice in the community, but some research suggests that trimethoprim should not be utilized empirically if resistance prevalence is greater than 20% [40,41]. The high prevalence of *E. coli* isolates resistant to trimethoprim in our study might indicate the need for revision of the empirical treatment guidelines for LTC residents in Australia. Overall, the rates of resistance observed in our study were similar to those reported in other Australian studies [28] and in the National Australian AMR surveillance [42] but lower than those reported in the UK [33]. These differences may be explained by different study periods, as our results show considerable changes over time for some antibiotics, differences in setting, differences in methodology, and differences in policies and guidelines guiding antibiotic utilization [28,30,32,33].

Our study has several strengths. We used an ecological study design and used routinely available data to compare antibiotic utilization data and resistance patterns at a population level. We were able to investigate a population of LTC residents in one region for which we had microbiology services provided by a single provider ensuring we had almost 100% capture of urine testing for LTC residents [43]. Furthermore, we quantified AMR concerning a clear reference population with a sensitive, specific, repeatable, and reproducible method of detection of resistance and a clear definition of the target bacteria [44]. However, our study also had limitations. First, due to the retrospective ecological study design, the comparison between the prevalence of AMR and antibiotic utilization was conducted at the population level, and our findings should be confirmed in future studies using individual resident-level data. However, previous research stated that antibiotic utilization at the population level might be more important in determining AMR risk in the community than at the individual level [45]. Secondly, our AMR patterns were based on routinely collected data, and research is needed to better understand when clinicians request microbiology testing to understand the role of microbiology testing in the management of UTI if testing is more likely to be performed for cases of antibiotic therapy failure, relapse, or reinfection [46,47], or if LTC residents receive empirical treatment with antibiotics without taking cultures [46,48,49]. Furthermore, we did not obtain any demographic information on the population from which the urine isolates were taken. Again, future individual-level studies are needed to explore the importance of including the dosing of antibiotics [50], the plausibility of multiple antibiotic and antimicrobial resistance combinations (co-selection of resistance) [33], and the presence of multidrug-resistant *E. coli*. Lastly, the lack of information on *E. coli* resistance to fosfomycin and nitrofurantoin hinders the global translatability of our findings. However, past research in Australian LTC facilities showed low utilization of these agents, with nitrofurantoin accounting for only 1% of all antibiotic treatment episodes (104 out of 10,460) and fosfomycin being utilized just once over a three-year period [20]. Given the low prevalence of *E. coli* resistance to fosfomycin and nitrofurantoin in countries where these agents are more frequently utilized [39], we suspect that *E. coli* resistance to these antibiotics is likely to be low in Australian LTC facilities.

## 4. Materials and Methods

### 4.1. Study Design

A retrospective, longitudinal ecological analysis of antibiotic utilization and *E. coli* AMR in urine isolates between 31 May 2016 and 31 December 2018 in LTC residents in the Illawarra Shoalhaven region of Australia was undertaken.

### 4.2. Study Setting

The Illawarra Shoalhaven region in New South Wales, Australia, lies south of Sydney. Unique to this area is that one pathology laboratory provides over 90% of pathology services to LTC facilities, providing a relatively complete microbiology dataset and a unique opportunity to understand AMR in the LTC setting [25].

### 4.3. Data Sources

#### 4.3.1. Microbiology Data

We obtained aggregated monthly data on antimicrobial susceptibility testing in urine *E. coli* isolates from LTC residents from the Illawarra Shoalhaven region conducted between May 2016 and December 2018 from the pathology laboratory [25]. Variables in the dataset included the date and susceptibility test result per antibiotic type.

The pathology laboratory used agar disc diffusion via the European Committee on Antimicrobial Susceptibility Testing (EUCAST) method, and isolates were classified as resistant (R), susceptible (S), or susceptible, increased exposure (I) [51]. For this study, isolates in the intermediate category were grouped with those in the susceptible category to follow the latest European Committee on Antimicrobial Susceptibility (EUCAST) definition [51]. Patient-level data (i.e., age and gender) were not provided.

#### 4.3.2. Antibiotic Utilization Data

Antibiotic utilization was determined using a pharmacy medication supply dataset from a longitudinal cohort of long-term care residents in the Illawarra Shoalhaven region. Residents who received one or more medicines during the study period were included in the dataset. The dataset provides each resident’s complete medication history and includes both prescription and non-prescription medicines. This dataset has been used to explore a range of research questions [52,53,54,55,56].

#### 4.3.3. Included Antibiotics

Standard urine microbiology panels in the Australian setting test for amoxicillin, trimethoprim, cephalexin, amoxicillin with clavulanic acid, norfloxacin, and nitrofurantoin [57]. In this analysis, data on the sensitivity to nitrofurantoin were not consistently reported in the microbiology dataset; therefore, nitrofurantoin was excluded from our analysis [58].

The Australian guidelines recommend the following antibiotics for the empirical treatment of urinary tract infection in the LTC setting: amoxicillin (ATC [59] J01CA04), amoxicillin with clavulanic acid (J01CR02), cefalexin (J01DB01), ciprofloxacin (J01MA02), fosfomycin (J01XX01), nitrofurantoin (J01XE01), norfloxacin (J01MA06), trimethoprim (J01EA01), and sulfamethoxazole with trimethoprim (J01EE01) [14]. As described above, nitrofurantoin was excluded from our analyses due to incomplete microbiology data collection. Fosfomycin was also excluded, as it is not routinely tested for in urine isolates and is rarely utilized, as it is not available via the national medicine reimbursement scheme [60].

Therefore, the antibiotics included in our analyses were amoxicillin (J01CA04), amoxicillin with clavulanic acid (J01CR02), cefalexin (J01DB01), ciprofloxacin (J01MA02), norfloxacin (J01MA06), trimethoprim (J01EA01), and sulfamethoxazole with trimethoprim (J01EE01).

### 4.4. Outcomes

Antimicrobial resistance was presented as the rate of resistant isolates per antibiotic type per 1000 *E. coli* urine isolates per month.

Antibiotic utilization was presented as the rate of antibiotic episodes per antibiotic type per 1000 residents per month. An antibiotic treatment episode was defined as an ongoing treatment episode with an included antibiotic. Breaks in treatment of less than 3 days were considered a single episode. Norfloxacin and ciprofloxacin utilization were combined, since the literature shows that decreased susceptibility of *E. coli* to one fluoroquinolone simultaneously reduces susceptibility to quinolones [50]. Trimethoprim and trimethoprim with sulfamethoxazole were combined, since both agents may contribute to the emergence of *E. coli* resistance to trimethoprim [61,62].

### 4.5. Variables and Analysis

Trends in the monthly prevalence of resistance and utilization were each modeled using a Generalized Additive Model, with a Poisson distribution, where the number of urine isolates and the number of LTC residents were included as offsets. The temporal relationship between urine isolate resistance and antibiotic utilization was compared visually.

All analyses were conducted using R (version 1.4.1106, R Foundation for Statistical Computing, Vienna, Austria).

## 5. Conclusions

This research shows a potential temporal relationship between antibiotic utilization and AMR in LTC over a short period of months. Both antibiotic utilization and AMR are dynamic, indicating that to influence AMR, up-to-date local antibiograms to inform antibiotic use could be a valuable tool in this setting.

## Figures and Tables

**Figure 1 antibiotics-14-00208-f001:**
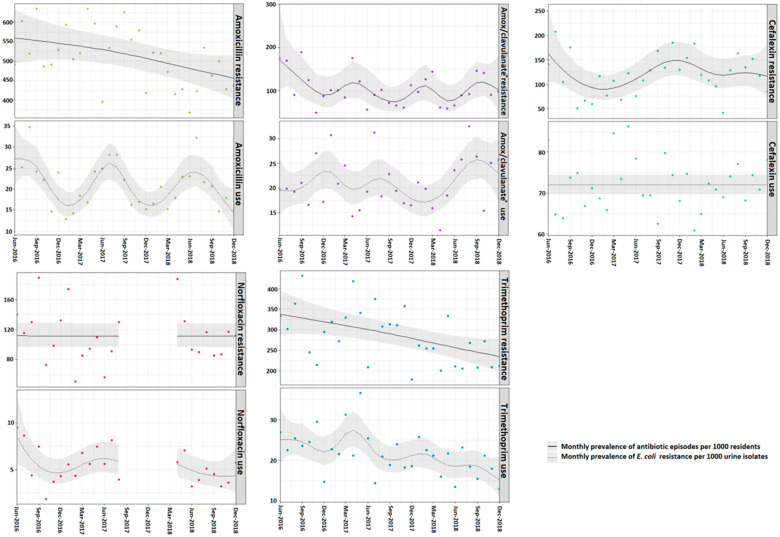
Monthly rate of systemic antibiotic episodes per 1000 residents and *E. coli* resistance per 1000 urine isolates. The grey lines indicate 95% confidence intervals around the monthly estimates. Norfloxacin utilization presents combined norfloxacin and ciprofloxacin treatment episodes. Trimethoprim utilization presents combined trimethoprim and trimethoprim with sulfamethoxazole treatment episodes. * Amoxicillin with clavulanate.

**Table 1 antibiotics-14-00208-t001:** The number of *E. coli* isolates resistant per antibiotic type and antibiotic treatment episodes utilized in LTCs between 31 May 2016 and 31 December 2018.

Antibiotic Type	Number of Urine *E. coli* Isolates Resistant (%) n = 10,835	Number of Antibiotic Treatment Episodes (%) n = 6795
Amoxicillin	1171 (51.3)	1007 (14.8)
Amoxicillin with clavulanic acid	237 (10.4)	1016 (15.0)
Cefalexin	276 (12.1)	3478 (51.2)
Norfloxacin	190 (11.1)	247 (3.6) ^1^
Trimethoprim	652 (28.6)	1047 (15.4) ^2^

^1^ Number of combined norfloxacin and ciprofloxacin treatment episodes. ^2^ Number of combined trimethoprim and trimethoprim with sulfamethoxazole treatment episodes.

## Data Availability

Restrictions apply to the availability of data used in this study. Data were obtained from Webstercare and Southern IML Pathology/WARRA and are available from the authors with permission from these providers.

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
