# Peer review of "Temporal Trends of Escherichia coli Antimicrobial Resistance and Antibiotic Utilization in Australian Long-Term Care Facilities"

_antibiotics, 2025, doi:10.3390/antibiotics14020208_

Round 1

Reviewer 1 Report

Comments and Suggestions for Authors

The article is thematically sound; however, its focus on a specific region constitutes a significant shortcoming. To address this, two options are available: either additional samples from diverse regions should be collected and included in the article, or they should be incorporated into the existing literature on the subject. 

Comments on the Quality of English Language

It is imperative that the language is rendered more fluent and comprehensible.

Author Response

Thank you for the opportunity to review our manuscript and to address the reviewers comments. We have undertaken a major revision of the manuscript. A point by point response to each reviewer comment is presented below.

  1. The article is thematically sound; however, its focus on a specific region constitutes a significant shortcoming. To address this, two options are available: either additional samples from diverse regions should be collected and included in the article, or they should be incorporated into the existing literature on the subject. 

Response 1: We agree that antimicrobial resistance (AMR) is a global issue. However, understanding AMR in local contexts, such as specific countries, regions or communities is an important step in understanding the global picture. Our previous research, exploring one health risk factors associated with E. coli resistance, highlighted the importance of local context in the identification of risk factors that contribute to AMR, with health care institutions settings such as long-term care identified as increasing the risk of E. coli resistance (Smit, C.C.H, et al. 2023. One Health Determinants of Escherichia coli Antimicrobial Resistance in Humans in the Community: An Umbrella Review. International Journal of Molecular Sciences). In the same research, we also highlighted significant diversity in terms of AMR risk factors demonstrating the need for in-depth studies in local contexts to understand context-specific drivers of AMR and allow the development of global strategies to reduce future AMR. In terms of global relevance, the Australian long-term care setting is considered comparable to that across North America, Europe and Oceania (Jain, B., Cheong, E., Bugeja, L. and Ibrahim, J., 2019. International Transferability of Research Evidence in Residential Long-term Care: A Comparative Analysis of Aged Care Systems in 7 Nations, Journal of the American Medical Directors Association) 20(12), pp.1558-1565.) highlighting the relevance of the research presented in this manuscript to the global residential aged care context.  Therefore this study provides valuable insights into real-life antibiotic utilization and antimicrobial resistance trends relevant to global residential aged care settings.  This global relevance of the setting has been clarified in the manuscript introduction.

  1. Comments on the Quality of English Language. It is imperative that the language is rendered more fluent and comprehensible.

Response 2: The English language throughout the manuscript has been revised to increase clarity and fluency.

Note Professors Keighley, Rogers, Robertson and Pont are all native English speakers.

Reviewer 2 Report

Comments and Suggestions for Authors

In the presented work, a study was conducted on the development of antibiotic resistance in E. coli with the content of antibiotics in urine, which are commonly used for this in medical institutions.

 The manuscript can be accepted for publication after major changes.

 1.     1. It is necessary to expand the introduction.

2.     2. There are not enough figures and tables in the description of the results, as well as a brief discussion of the results. It would probably be useful to add a comparison of drug consumption in different countries, in animal husbandry, etc.

3.     3. Can the authors suggest a relationship between the structure of an antibiotic and the development of resistance?

4.     4. Does the development of resistance depend on the concentrations of medicinal compounds used or only on the duration of use?

Author Response

Thank you for the opportunity to review our manuscript and to address the reviewers comments. We have undertaken a major revision of the manuscript. A point by point response to each reviewer comment is presented below.

Comments and Suggestions for Authors

In the presented work, a study was conducted on the development of antibiotic resistance in E. coli with the content of antibiotics in urine, which are commonly used for this in medical institutions.

The manuscript can be accepted for publication after major changes.

  1. It is necessary to expand the introduction.

Response 1: We have expanded the introduction to include a discussion of the mechanisms of resistance per antibiotic type and an explanation for the choice and relevance of the study setting in the introduction.

  1. There are not enough figures and tables in the description of the results, as well as a brief discussion of the results. It would probably be useful to add a comparison of drug consumption in different countries, in animal husbandry, etc.

Response 2: Thank you for this comment. We agree that a one-health approach and global perspective on the topic of antimicrobial resistance is important, however, the scope of this research was antibiotic consumption and AMR in the residential aged care setting. As discussed previously, understanding AMR in local contexts, such as specific countries, regions, or communities, is an important part of understanding the global picture. We also agree that future research exploring antibiotic consumption and antimicrobial resistance integrated across all one health contexts, that is, environmental, agricultural and human contexts, is needed, however, as identified in the Umbrella review into risk factors for AMR previously published by our team (Smit, C.C.H, et al. 2023. One Health Determinants of Escherichia coli Antimicrobial Resistance in Humans in the Community: An Umbrella Review. International Journal of Molecular Sciences), research across the majority of these contexts is limited and currently, data to allow comparison in different contexts is lacking.

  1. Can the authors suggest a relationship between the structure of an antibiotic and the development of resistance?

Response 4: The transition of commensal E. coli to pathogenic antimicrobial-resistant E. coli is proposed to be primarily driven by horizontal gene transfer of resistant genes or mobile genetic elements. The mechanism of resistance varies per antibiotic type. Acquisition of TEM-1/TEM-2- β-lactamase is known for causing resistance to amoxicillin, whereas extended-spectrum β-lactamase leads to resistance of E. coli to amoxicillin with clavulanic acid and cefalexin. Trimethoprim resistance is established through mutations of sul and dfr genes and gyrA/parC and gyrB/pare are responsible for resistance to quinolones. This has been added to the introduction of the manuscript.

  1. Does the development of resistance depend on the concentrations of medicinal compounds used or only on the duration of use?

Response 4: Thank you for this useful comment. In this study we were unable to consider antibiotic strength, dose or treatment duration, however, we agree this is an important area for future research and we have added this as a limitation to the manuscript.

Reviewer 3 Report

Comments and Suggestions for Authors

The sampling method was convenient  sampling  (non probability), therefore how a prevalence was calculated should be elaborated.

The study duration is 2.5 years and monthly data are collected, is this enough for trends analysis? pls justify.

The methodology section needs to be elaborated.

Author Response

Thank you for the opportunity to review our manuscript and to address the reviewers comments. We have undertaken a major revision of the manuscript. A point by point response to each reviewer comment is presented below.

  1. The sampling method was convenient  sampling  (non probability), therefore how a prevalence was calculated should be elaborated.

Response 1: Thank you for your comment. The data presented are rates not prevalence and the manuscript text has been revised accordingly.

The study duration is 2.5 years and monthly data are collected, is this enough for trends analysis? pls justify.

Response 2: This study aimed to explore potential trends between antibiotic consumption and AMR using data from 36 time points. While we chose a visual rather than a statistical analytical approach, 36 data points provide a sample size that is well in excess of the 16 to 17 data points generally accepted as the minimum sample size for statistically modeling temporal trends. (Hyndman, R.J. and Kostenko, A.V., 2007. Minimum sample size requirements for seasonal forecasting models. foresight,)

  1. The methodology section needs to be elaborated.

Response 3: Thank you for your comment. We have revised the methods section.

Reviewer 4 Report

Comments and Suggestions for Authors

Please Add this reference to your study :

Five-Year Antimicrobial Resistance Patterns of Urinary Escherichia coli at an Australian Tertiary Hospital: Time Series Analyses of Prevalence Data | PLOS ONE

10.1371/journal.pone.0164306 

Excellent study

Comments on the Quality of English Language

Excellent  manuscript

Author Response

Please Add this reference to your study :

Five-Year Antimicrobial Resistance Patterns of Urinary Escherichia coli at an Australian Tertiary Hospital: Time Series Analyses of Prevalence Data | PLOS ONE

10.1371/journal.pone.0164306 

Excellent study

Response: Thank you for your kind words regarding our study. We have included the reference in our manuscript.

Round 2

Reviewer 1 Report

Comments and Suggestions for Authors

Acceptable in the light of the changes made

Reviewer 2 Report

Comments and Suggestions for Authors

The manuscript may be accepted for publication.

Reviewer 3 Report

Comments and Suggestions for Authors

Thank you for taking time to do the amendments. The changes are satisfactory, the article can be accepted in its current version.